# Excess forest mortality is consistently linked to drought across Europe

Cornelius Senf[1✉], Allan Buras[2], Christian S. Zang[2], Anja Rammig[2] & Rupert Seidl [1,3]

Pulses of tree mortality caused by drought have been reported recently in forests around the globe, but large-scale quantitative evidence is lacking for Europe. Analyzing high-resolution annual satellite-based canopy mortality maps from 1987 to 2016 we here show that excess forest mortality (i.e., canopy mortality exceeding the long-term mortality trend) is significantly related to drought across continental Europe. The relationship between water availability and mortality showed threshold behavior, with excess mortality increasing steeply when the integrated climatic water balance from March to July fell below −1.6 standard deviations of its long-term average. For −3.0 standard deviations the probability of excess canopy mortality was 91.6% (83.8–97.5%). Overall, drought caused approximately 500,000 ha of excess forest mortality between 1987 and 2016 in Europe. We here provide evidence that drought is an important driver of tree mortality at the continental scale, and suggest that a future increase in drought could trigger widespread tree mortality in Europe.

[1] Ecosystem Dynamics and Forest Management Group, Technical University of Munich, Hans-Carl-von-Carlowitz-Platz 2, 85354 Freising, Germany. [2] Land Surface-Atmosphere Interactions, Technical University of Munich, Hans-Carl-von-Carlowitz-Platz 2, 85354 Freising, Germany. [3] Berchtesgaden National Park, Doktorberg 6, 83471 Berchtesgaden, Germany. ✉email: cornelius.senf@tum.de

Accumulating evidence suggests that hotter drought is increasingly causing pulses of tree mortality throughout the globe[1], substantially impacting forest ecosystem functions and services[2]. While drought can impair photosynthesis and directly trigger mortality through carbon starvation and hydraulic failure[3], it is most often the combined effect of drought and secondary mortality agents that causes large-scale pulses of tree mortality[4–6]. In particular, hotter droughts are an inciting factor of tree mortality from insects[6–11] and may further increase the size and intensity of forest fires[12–15]. As droughts are predicted to become more frequent and more intense under climate change[16,17], concerns have been raised whether forest ecosystems might become increasingly susceptible to drought-induced ecosystem collapse[18–20]. However, notwithstanding the growing number of detailed physiological observations at the local scale, the quantitative understanding of drought-induced tree mortality across large spatial scales remains limited.

Across Europe, several local studies have reported increased tree mortality in response to drought recently[10,21–28]. Those studies have been mostly confined to one particular ecosystem or few species (e.g., Scots pine in the Swiss Alps as well as in lowland Germany[10,28], Oak in Poland[26], forests of the Iberian Peninsula[21]) and thus cannot be generalized across Europe. Many studies are furthermore limited to relatively short time series (e.g., 12 years of data for one of the few existing pan-European studies on drought mortality[22]) or they focus on one particular drought event (e.g., the Iberian drought of 1994[24,25], the European heatwave of 2003[29]), which limits analyses across time. Finally, many forests in Europe are intensively managed, and a drought-related decrease in tree vitality often results in an increase in harvest activity (i.e., sanitation logging) before large-scale dieback of trees can occur. Given the patchy evidence and the lack of broad-scale assessments of drought-related tree mortality, the role of drought as an agent of tree mortality across Europe remains unclear.

We here present a systematic and quantitative analysis of drought-related tree mortality over 30 years and across all of continental Europe's forests. Continental Europe here includes all countries of the European continent larger than 10,000 km², excluding Russia (a total of 35 countries with a forest area of more than 210 million ha). Our research addresses the following questions: (1) Is tree mortality statistically related to drought across Europe's forests? (2) Is the relationship between mortality and water availability linear, or does it show threshold behavior? (3) Where and when did hotspots of drought-related mortality occur between 1987 and 2016? (4) What was the total excess forest mortality (i.e., forest mortality exceeding the long-term mortality trend) caused by drought in this period? To answer these questions we use high-resolution maps of forest canopy mortality (including natural causes of tree death as well as harvesting by humans) derived from the full Landsat archive[30]. Using these maps we calculate the annual fractional deviation in forest canopy mortality from the long-term trend at a spatial grain of 0.5 degrees: A value around zero thus indicates no deviation in canopy mortality from the long-term trend, while values greater than zero indicate excess canopy mortality, and values below zero indicate an annual canopy mortality deficit (see Supplementary Figs. 1 and 2). We use generalized additive mixed modeling to test whether the fractional deviation in forest canopy mortality was statistically related to the climatic water balance (CWB; the difference between the monthly precipitation sum and the monthly potential evapotranspiration) of each grid cell and year. We hypothesize a consistent negative relationship between CWB and the fractional change in forest canopy mortality, that is with decreasing water availability excess mortality is expected. Using predictions from the model, we further investigate whether the probability of a grid cell to experience excess

forest canopy mortality increases linearly with decreasing CWB, or whether the relationship shows threshold behavior. Finally, we identify hotspots of drought-related excess forest canopy mortality, that is regions and years where excess mortality co-occurred with drought, and estimate the total excess mortality in response to drought across continental Europe over the period 1987 to 2016. We find that excess forest canopy mortality is statistically linked to drought and that drought caused approximately 500,000 ha of excess forest mortality between 1987 and 2016 in Europe. Drought is thus an important driver of tree mortality at the continental scale.

## Results

We found strong evidence for a negative relationship between CWB and excess forest canopy mortality. The relationship was consistent across Europe and indicated that forest canopy mortality increased significantly with drier conditions (Table 1). The relationship was strongest using CWB values averaged from March to July (Supplementary Fig. 3), and we thus report results for this integration period throughout the text. We found strong evidence for a non-linear effect of water availability on tree mortality (Supplementary Table 1). Specifically, the predictive performance of our model substantially increased when accounting for possible non-linear relationships between CWB and excess canopy mortality. The final model was substantially better than a null model based solely on random spatial and temporal variability (Supplementary Table 1) and explained 6% of the pan-European variability in forest canopy mortality.

We calculated the probability of excess forest canopy mortality for different CWB levels by creating random draws from the model (Fig. 1). The probability of excess mortality (i.e., excess mortality >0%) remained relatively stable for CWB values of up to −1.6 standard deviations (SD; 95% credible interval: −2.0– −1.3), but increased steeply with more negative CWB values (for details on the thresholds determination see Supplementary Fig. 7). More specifically, for a CWB of −1 SD relative to the long-term average, the probability of a grid cell experiencing excess mortality was 43.0 (33.3–56.2)%. The probability increased to 55.9 (45.3–67.7)% for a CWB of −2 SD, and to 91.6 (83.8–97.5)% for a CWB of −3 SD (Fig. 1). This threshold behavior was consistent for different levels of excess mortality (Fig. 1), i.e., we also observed a sharp increase in the probability of a grid cell experiencing >25%, >50% or >100% more mortality than the long-term trend would suggest. The probability of experiencing no excess mortality, that is average or below average mortality rates (blue line in Fig. 1), sharply decreased once the CWB threshold of −1.6 SD was crossed.

Through a spatial overlay of excess forest canopy mortality maps (Supplementary Fig. 1) and gridded CWB data we identified hotspots of drought-related excess mortality across Europe between 1987 and 2016 (Fig. 2). Hotspots were defined as regions and years where excess forest canopy mortality coincided with CWB values smaller than −1.6 SD of a grid cell's long-term average (thus below the threshold identified above). Maps using lower and upper threshold bounds are shown in Supplementary Figs. 8 and 9. Hotspots of drought-related excess canopy mortality were identified for 1992 in central-eastern Europe, 1994 in Spain, 1999 and 2002 in the Baltic states, 2000 in south-eastern Europe, 2003 in central and western Europe, 2005 in the Iberian Peninsula, 2006 in central-eastern Europe as well as Finland, and 2007 and 2012 in south-eastern and eastern Europe, among others.

From the maps shown in Fig. 2 we estimated a total of 511,059 ha (152,657 and 979,775 ha for the lower and upper CWB threshold, respectively) of excess forest canopy mortality in response to drought over the period 1987 to 2016 (Fig. 3). This

**Table 1 Estimates for all parameters of the final generalized additive mixed model, testing for a consistent link between drought and canopy mortality.**

| Parameter | Mean | 95% credible interval | $\hat{R}$ |
|---|---|---|---|
| Smooth-term: | | | |
| Degree of smoothing ($\sigma^2_{smooth}$) | 0.51 | 0.27–0.96 | 1.00 |
| Group-level parameters | | | |
| Spatial variability in intercept ($\sigma^2_{\alpha_i}$) | 0.18 | 0.17–0.18 | 1.00 |
| Spatial variability in slope ($\sigma^2_{\beta_i}$) | 0.03 | 0.02–0.03 | 1.00 |
| Correlation between spatial variability in intercept and slope ($\rho\,\sigma^2_{\alpha_i}\sigma^2_{\beta_i}$) | 0.03 | −0.05–0.12 | 1.00 |
| Temporal variability in intercept ($\sigma^2_{\alpha_t}$) | 0.08 | 0.06–0.10 | 1.00 |
| Temporal variability in slope ($\sigma^2_{\beta_t}$) | 0.04 | 0.03–0.06 | 1.00 |
| Correlation between temporal variability in intercept and slope ($\rho\,\sigma^2_{\alpha_t}\sigma^2_{\beta_t}$) | −0.08 | −0.44–0.29 | 1.00 |
| Population-level parameters | | | |
| Intercept ($\alpha_0$) | 0.01 | −0.02–0.04 | 1.01 |
| Slope ($\beta_0$) | −1.88 | −2.80–−0.97 | 1.00 |
| Family-specific parameters | | | |
| Residual variance ($\sigma^2_{mort}$) | 0.13 | 0.13–0.14 | 1.00 |
| Mixing parameter ($\lambda$) | 0.53 | 0.53–0.53 | 1.00 |

The model includes a smooth term, allowing for a non-linear response of mortality to drought. Shown are the mean of each parameter's posterior distribution and the central 95% credible interval. We further report the Gelman-Rubin statistic ($\hat{R}$), with values close to one indicating proper convergence of the joint posterior. Parameters are grouped into smooth term parameters (i.e., degree of smoothing), group-level parameters (i.e., variability of both intercept and slope in space [i.e., grid cells $i$] and time [i.e., years $t$]), population-level parameters (i.e., population averages after accounting for spatial and temporal variability), and family-specific parameters. See Supplementary Fig. 4 for a visualization of the non-linear response. See Supplementary Figs. 5 and 6 for spatial and temporal variability in slopes ($\beta_i$ and $\beta_t$).

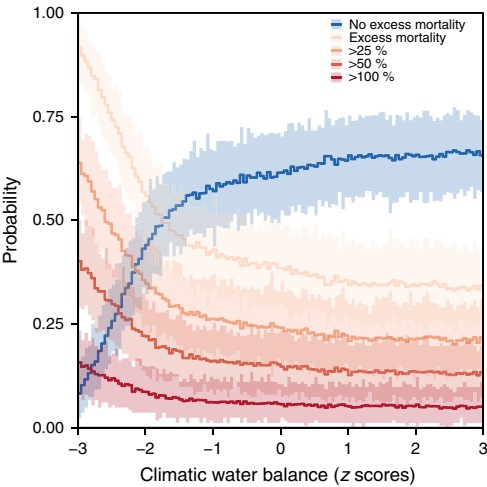

**Fig. 1 The probability of excess forest canopy mortality in Europe's forests related to the climatic water balance (CWB).** Values for different levels of mortality above the values expected from the long-term mortality trend are shown, with 100% excess mortality indicating a doubling of the annual area of canopy mortality. CWB is expressed as z-score, that is 0 indicates average CWB and ±1 indicate ±1 standard deviation below/above the average. Lines present the average estimate derived from $n = 8{,}000$ random draws of the full model, that is including parameter and model uncertainty. Ribbons represent the 95% credible interval derived by splitting all random draws into n = 100 bins and repeating the calculation for each bin.

amount of excess mortality equals approximately 1.4% (0.4–2.7% for the lower and upper CWB threshold, respectively) of all the canopy mortality (from natural as well as human causes) recorded for the same period across Europe. The share of total canopy mortality that was related to drought varied widely (Fig. 4), with drought related mortality accounting for 30% or more of the total forest canopy mortality in the past three decades in some regions (e.g., parts of the Iberian Peninsula, France and eastern Europe). Drought-related excess forest canopy mortality moreover occurred in pulses (Fig. 3), with the five largest pulses in the years 2005 (excess mortality of 127,688 ha), 2003 (103,311 ha), 2006 (48,306

ha), 2000 (34,026 ha), and 1994 (33,075 ha). Thus, four out of the five largest drought-related forest diebacks in Europe in the past three decades occurred in the 21st century.

## Discussion

We here present a systematic quantification of the relationship between tree mortality and drought for continental Europe, showing that low water availability causes considerable levels of excess forest canopy mortality. Our findings are in line with a growing body of literature highlighting the importance of drought for forest ecosystems across the globe[1,5,12]. Yet, our analyses add additional critical information, because we here identified a continentally consistent threshold value beyond which drought-related forest mortality is likely to occur. We further identified several hotspots of drought related excess canopy mortality across Europe, with most of them occurring after the year 2000, suggesting an increased occurrence of global change type droughts in Europe. The hotspots identified herein correspond well with local reports across Europe[10,21–27] and reflect well-described large-scale drought events[31], such as the European heatwave of 2003[29]. We, however, also identified several hotspots that are not well documented in the literature, in particular in eastern and southeastern Europe, indicating that those regions need greater attention in the context of drought-related forest mortality research.

Some regions in Europe were affected more frequently by drought-related mortality than others, such as the Iberian Peninsula or south-eastern Europe (see also Supplementary Fig. 5). Those regions are characterized by already dry and warm climates, and correspond to the range margins of common European tree species such as European beech, Scots Pine, or European Oak. Those regions might thus be particularly vulnerable to drought-related tree mortality[19,32]. Broad biogeographic patterns can, however, be modulated by factors influencing drought susceptibility at the local scale, with microclimate overriding coarser climate filters (i.e., south-exposed slopes and dry valleys in the Alps[33]) and structural legacies determining spatial patterns of forest dieback[34]. We further found that the effect of drought on canopy mortality was non-linear and showed threshold behavior. This suggests that Europe's forest ecosystems are generally well adapted to variability in water availability, but respond with

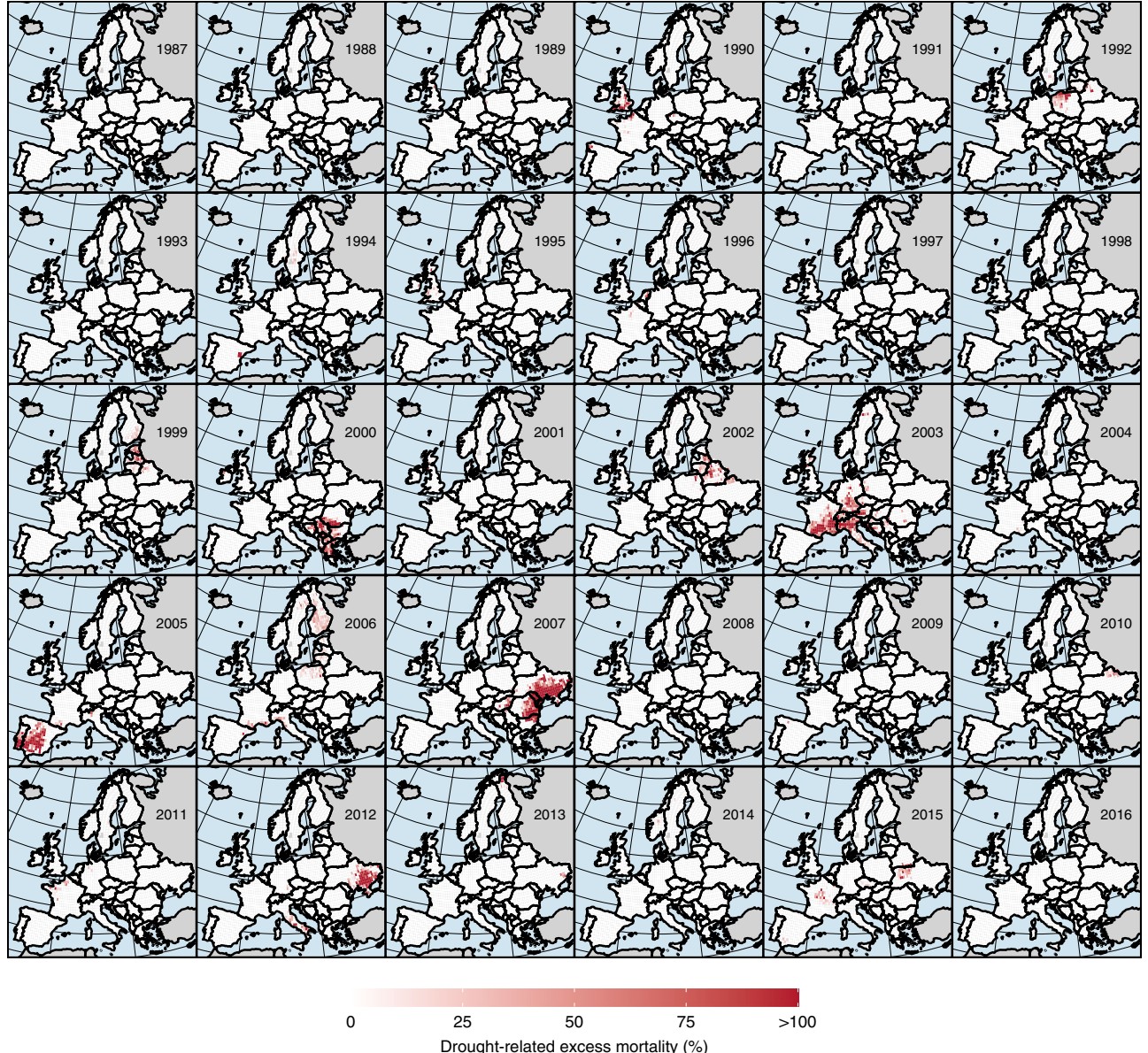

**Fig. 2 Hotspots of excess forest canopy mortality in response to drought.** Hotspots were defined as regions and years where excess forest canopy mortality coincided with CWB values smaller than −1.6 standard deviations below the local average. See Supplementary Figs. 8 and 9 for hotspot maps using lower and upper bound thresholds. Note that we here report relative changes, whereas absolute estimates in terms of forest area are given in Fig. 3. Background maps are from https://naturalearthdata.com. The map was created by C. Senf.

increased mortality once a certain threshold is crossed. While we identified a pan-European CWB threshold for excess canopy mortality at −1.6 SD below the long-term average (corresponding to the 4% driest conditions recorded in Europe in the past three decades), we note that this threshold might vary by region and species, depending on region-specific climate variability of the past and drought tolerance of local tree species[35]. The threshold behavior emerging from our data is well in line with previous research and current process-understanding of tree responses to drought[36,37]. It also suggests that under increasing drought intensity[38] and hotter droughts under climate change[36], tree mortality could increase disproportionally. In this respect we note that the effect of drought on tree mortality identified here was strongest for droughts lasting over several months (March to July), suggesting that an increase in drought duration – as expected for droughts under climate change[39] – might further amplify tree mortality in the future.

While droughts in general and hotter droughts in particular can trigger mortality directly, it is most likely the combined effect of drought and secondary mortality agents that led to the tree mortality hotspots identified in this study. One of the most important mortality agents in terms of hotter droughts is fire[12–15]. For example, the year 2005 marks one of the most severe fire seasons on the Iberian Peninsula in recent decades[40], explaining the significant hotspot of drought-related tree mortality identified in our analysis in the same year across the western Iberian peninsula (Fig. 2). Likewise, Ukraine and neighboring countries were hit by widespread forest fires during the drought of 2007 and 2011/2012, as were the Balkan peninsula in 2000 and the Baltic states including Belarus in 2002[41], all of which are well visible in our hotspot assessment (Fig. 2). Overall, our estimates of drought related excess mortality correlate reasonably well with the total area burned reported by countries (Pearson $r = 0.43$ [0.33 and 0.52 for the lower and upper CWB threshold,

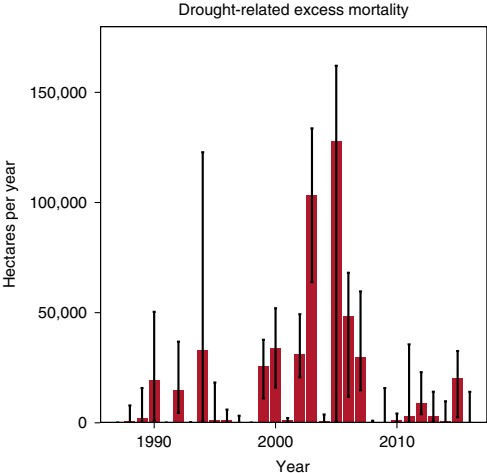

**Fig. 3 Forest area affected by drought-related excess forest canopy mortality between 1987 and 2016 in Europe.** The red bars sum all excess forest canopy mortality across the $n = 2{,}913$ grid cells that co-occurred with CWB values smaller $-1.6$ standard deviation below the local average. The error-bars sum all excess mortality that co-occurred with CWB values smaller than $-1.3$ (upper bound) and $-2.0$ (lower bound) standard deviation below the local average, respectively. The error-bars thus show the potential range of drought-related excess mortality depending on different thresholds used for defining a drought that caused excess forest canopy mortality.

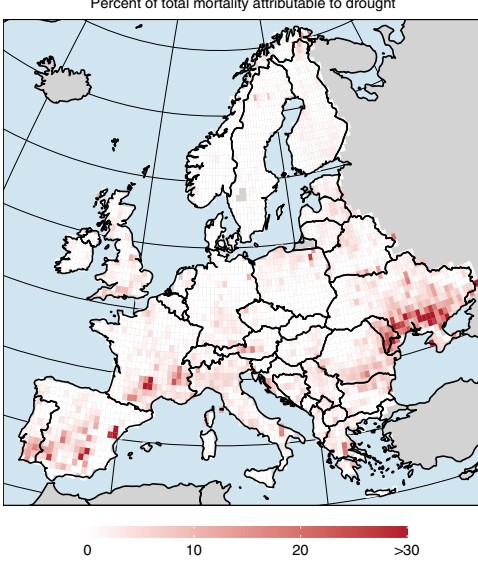

**Fig. 4 Percent of the total forest canopy mortality attributable to drought-related excess forest canopy mortality between 1987 and 2016 across Europe.** A value of 30 means that 30% of the total forest canopy mortality in this particular grid cell and over the whole period was drought-related excess forest canopy mortality given the definitions used in Fig. 2. See Supplementary Figs. 10 and 11 for maps using lower and upper bound thresholds. Background maps are from https://naturalearthdata.com. The map was created by C. Senf.

respectively]; Supplementary Fig. 12), indicating that a significant proportion of the drought-related excess mortality identified in this study may be caused by fire disturbances. Hotter droughts also are an inciting factor of tree mortality from insects[6–11]. For example, during the 2003 heatwave in Europe, mortality from several insect species increased in direct response to the increasing heat and lack of water in France[11], supporting

the widespread dieback identified in our analysis (see Fig. 2 for year 2003). However, we like to note that while bark beetle mortality (mostly by *Ips typographus*) was linked to drought in past studies[11,42–44], there is often a temporal lag of one to three years between drought occurrence and the detection of dead trees in satellite products[14,42]. Hence, our analysis likely does not include drought-related mortality by bark beetles, which contributed significantly to tree mortality in recent years across Central Europe[45]. In contrast to indirect mortality by fire or insects, there are also well-documented instances of tree dieback as direct consequence of drought. For example, in Spain in 1994 widespread dieback of Oak and Pine was caused by drought[21], which might explain the small but distinct hotspot identified in Spain in this year (Fig. 2), but we note that 1994 also constitutes a severe fire year (Supplementary Fig. 12).

We here provide an estimate of the total excess forest canopy mortality that has occurred in Europe's forests in response to drought, which totals to more than half a million ha over the past 30 years. This corresponds to approximately 1.4% of all canopy mortality recorded in Europe in the past three decades, and is thus smaller but in the same order of magnitude as mortality from other major natural disturbance agents in Europe[46]. While this number highlights that drought is an important but likely less frequent agent of tree mortality across Europe, important context information needs to be considered in the interpretation of our results. First, the satellite data used here pose limitations with regard to the interpretability of results. For instance, we did not identify causal mortality agents, meaning that excess mortality could be caused by any agent co-occurring with drought, such as fire, and even those mechanistically unrelated to drought (e.g., windthrow coincidentally cooccurring with drought). The advantage of our approach is, however, that it guarantees inclusion of all excess mortality triggered by drought, including indirect effects from, e.g., increased sanitation logging in response to drought. Such effects would be neglected if an a priori categorization of canopy mortality would have been performed. Second, our results are based on validated maps of canopy mortality and not – as often used – maps of anomalies in vegetation indices without verified skills to detect mortality events[47,48]. Our results thus report actual mortality events and not only changes in photosynthetic activity, which might be ephemeral. Third, we here defined drought as the negative deviation from average water availability, which is a pragmatic but limited view on drought[49]. We thus acknowledge that the local effects of drought on tree mortality can be considerably more complex than continental-scale relationships reported herein suggest. These limitations notwithstanding, we here provide systematic continental-scale evidence that drought is an important agent of tree mortality in Europe. Our results suggest that an increase in the frequency and severity of drought under climate change, that is an increase of hotter drought[36], could lead to substantial tree mortality across Europe's forests, as has been already observed in 2018 and 2019[45,50].

## Methods
**Identifying tree mortality anomalies.** We identified anomalies in tree mortality from an existing high-resolution canopy mortality map created from Landsat satellite data[51]. While the data are described in detail in Senf and Seidl 2020[30], we here provide the salient details necessary for understanding our approach: The map has a spatial grain of 30 m and is based on a supervised classification of Landsat satellite image time series. The map indicates for each pixel whether a canopy mortality event has occurred between 1987 and 2016 and if so, in which year it occurred. The formal agent of the canopy mortality event (i.e., natural or human cause) is not classified. The map also identifies all pixels that have been forested over the same time period. The overall accuracy of the map is 92%, with a commission error of 15% and an omission error of 33% for detecting canopy mortality events[30]. Consequently, the map is conservative, omitting some disturbances in favor of not predicting false canopy mortality events.

We spatially overlaid the canopy mortality map with a 0.5-degree grid (total of 3,113 grid cells) and spatially aggregated the disturbance maps to the annual sum of 30 m pixels indicating canopy mortality (i.e., the annual total area of canopy mortality recorded within a 0.5-degree grid cell). We also aggregated the total forest area to the 0.5° grid cell level, allowing us to derive an annual mortality rate per grid cell (total annual area of canopy mortality divided by the total forest area). We excluded all cells that consisted mostly of non-forest ecoregions according to Olsen et al.[52] (e.g., tundra and grassland ecoregions in northern Scandinavia), which resulted in the exclusion of 200 grid cells (6%). Based on previous research we expected the average mortality rate to increase across Europe[53,54]. However, in some years there can be substantially higher mortality rates than what would be expected even under a long-term increasing trend, a fact that we here define as excess mortality. To quantify excess mortality for each year and grid cell, we used an established modeling framework based on logistic regression (described in Senf et al.[53]) to model each cell's long-term trend in canopy mortality. In essence, the model predicts the annual proportion of pixels with canopy mortality over the total number of forested pixels using a logistic model with binomial error distribution and time as predictor, modeling the change in the average proportion of forest area subject to canopy mortality (i.e., the mortality rate) over time (see trend line in Supplementary Fig. 3). We subsequently derived the residuals from the model and normalized them by the annual fitted values from the trend line. The resultant value indicates the annual fractional deviation in mortality from the long-term trend: A value close to zero indicates no change in mortality compared to the long-term trend, values greater than zero indicate excess mortality, and values smaller than zero indicate a mortality deficit (see Supplementary Fig. 1).

**Drought indices**. We focused on the climatic water balance (CWB) as index of drought. While there is a multitude of drought indices[55], we here used the CWB as it is commonly used across Europe[38,56] and has been shown to correlate well with tree mortality at the landscape scale[42]. The climatic water balance was defined as the difference between the monthly precipitation sum and the monthly potential evapotranspiration, which was obtained from the mean monthly temperature via the Thornthwaite equation[57]. The base data (precipitation and temperature) was derived from CRU TS 4.03[58] at a spatial resolution of 0.5 degrees. To account for different drought durations, we calculated average values over one to six months, left-centered on each month from March to August. The final dataset thus included monthly CWB observations from March to August with variable temporal integration periods. We z-transformed each observation by subtracting the mean and dividing by the standard deviation of the full time series. The CWB is thus expressed as the relative anomaly compared to the grid cell average.

**Statistical analysis**. To test for a robust statistical relationship between tree mortality and drought we used a mixed effect model to predict the annual fractional deviation in mortality from the long-term trend, $\text{mort}_{it}$, for grid cell $i$ and year $t$ from the CWB value of cell $i$ and year $t$:

$$\text{mort}_{it} = (\alpha_0 + \alpha_i + \alpha_t) + (\beta_0 + \beta_i + \beta_t) * \text{CWB}_{it} + \varepsilon_{\text{mort}} \quad (1)$$

$$\varepsilon_{\text{mort}} \sim \text{N}(0, \sigma_{\text{mort}}^2) \quad (2)$$

$$\binom{\alpha_i}{\beta_i} \sim \text{MVN}(0, \Sigma_i); \text{ with } \Sigma_i = \begin{pmatrix} \sigma_{\alpha_i}^2 & \rho \, \sigma_{\alpha_i}^2 \sigma_{\beta_i}^2 \\ \rho \, \sigma_{\alpha_i}^2 \sigma_{\beta_i}^2 & \sigma_{\beta_i}^2 \end{pmatrix} \quad (3)$$

$$\binom{\alpha_t}{\beta_t} \sim \text{MVN}(0, \Sigma_t); \text{ with } \Sigma_t = \begin{pmatrix} \sigma_{\alpha_t}^2 & \rho \, \sigma_{\alpha_t}^2 \sigma_{\beta_t}^2 \\ \rho \, \sigma_{\alpha_t}^2 \sigma_{\beta_t}^2 & \sigma_{\beta_t}^2 \end{pmatrix} \quad (4)$$

The model allows for random variation in both the intercept (average $\text{mort}_{it}$) and slope (effect of CWB on $\text{mort}_{it}$) among grid cells and years. The random variations in intercept and slope are modeled using a zero-centered multivariate normal distribution (MVN) with variance-covariance matrix $\Sigma_i$ and $\Sigma_t$, respectively. For the model described above, we evaluated which combination of observation month (i.e., March to August) and integration period (one to six months) best fit our data. We fitted one model for each combination of observation month and integration period using maximum likelihood methods implemented in the lme4 package[59]. We then compared the model performance among all combinations using Akaike's Information Criteria (AIC), and chose the observation month and integration period which yielded the smallest AIC value. The final model, however, showed some deviations of the model residuals from the normality assumption of a Gaussian error distribution. We therefore re-fitted the best combination of observation month and integration period using Bayesian methods using an exponentially modified Gaussian error distribution: $\varepsilon_m = \text{exGaussian}(0, \sigma_{\text{mort}}^2, \lambda)$. In essence, the exponentially modified Gaussian distribution is a mixture of a normal Gaussian and an exponential distribution, where $\lambda$ is the rate of the exponential distribution. Using an exponentially modified Gaussian error distribution instead of a normal Gaussian error distribution substantially improved the behavior of the model residuals and substantially reduced the influence of a few very high excess mortality values on the model estimates. Bayesian inference was done using Monte-Carlo-Markov-Chain (MCMC) methods implemented in the software Stan[60], accessed via the brms

package[61] in the statistical software R[62]. We ran four chains à 4,000 samples (of which 2,000 were warm-up) to guarantee robust convergence of the joint posteriors. We used weakly regularizing priors as implemented in the brms package for all priors. We checked the convergence of the MCMC chains using trace plots and the Gelman-Rubin statistic (R̂), which compares the within-chain variability to the between-chain variability over all MCMC chains and should approach one if all chains converge to a similar solution[63].

We finally compared three model set-ups: First, we fitted a null-model containing only an intercept but the similar random effect structure as the models described above. That is, the intercept was allowed to vary randomly among grid cells and years following $\alpha_i \sim \text{N}(0, \sigma_{\alpha_i}^2)$ and $\alpha_t \sim \text{N}(0, \sigma_{\alpha_t}^2)$. The null-model thus assumes that anomalies in tree mortality emerge from pure stochasticity in time and space. Second, we fitted the full model as described above, assuming a linear effect of CWB on the annual fractional deviation in mortality. Third, we added a smoothing term to the effect of CWB on the annual fractional deviation in mortality:

$$\text{mort}_{it} = (\alpha_0 + \alpha_i + \alpha_t) + (\beta_0 + \beta_i + \beta_t) * \text{CWB}_{it} + \text{f}(\text{CWB}_{it}) + \varepsilon_m \quad (5)$$

$$\text{f}(\text{CWB}_{it}) = \sum_{j=1}^{k} \text{B}_j(\text{CWB}_{it})\gamma_j \text{ with } k = 10 \quad (6)$$

with $\text{B}_j$ being a thin plate spline basis function[64] and $\gamma_j \sim \text{N}(0, \sigma_{\text{smooth}}^2)$ the weight of each basis function. The smoothing term allows for a non-linear response of canopy mortality to changes in the CWB. This was done to identify a potential threshold in the effect of drought on canopy mortality, e.g., a disproportional increase in excess canopy mortality beyond a certain drought severity. The variance component of $\gamma_j$ regularizes the degree of smoothing and is modeled as hierarchical hyper-parameter, that is the degree of smoothing is determined from the data with regularization from the priors implemented in the brms package[65]. We compared all three models by their approximate leave-one-out cross-validated expected log predictive density (LOO-ELPD), which is an information-theoretic criterion of the model fit for Bayesian models[66] similar to the AIC. The final model (i.e., with or without smoothing function) was chosen based on the maximum LOO-ELPD.

To visualize the effect of CWB on excess mortality—and to assess potential non-linearities and threshold—we created random posterior draws from our model (i.e., including both parameter and model uncertainty) over a range of CWB values. From the posterior draws we calculated the probability of a grid cell/year to experience excess mortality at different excess mortality levels (>0%, >25%, >50%, >100% excess mortality), and plotted these over CWB in order to detect potential thresholds in the response of tree mortality to changing water availability. For determining thresholds, we used an established breakpoint detection algorithm[67] that splits a time series into subsets of approximate linear change. We set the maximum number of breakpoints to one and ran the breakpoint algorithm for each draw of the posterior, allowing us to identify the uncertainty in potential threshold values.

**Hotspots mapping and area-estimates**. To identify hotspots of drought-related excess canopy mortality we overlaid the map of fractional deviation in canopy mortality (Supplementary Fig. 1) with gridded CWB data and identified all excess canopy mortality occurring under drought condition. Drought conditions were defined via the CWB threshold (and its uncertainty) identified in the breakpoint analysis described above.

**Reporting summary**. Further information on research design is available in the Nature Research Reporting Summary linked to this article.

## Data availability
All data are stored in a permanent repository under the following link: https://doi.org/10.5281/zenodo.3924656.

## Code availability
All code is available under: https://github.com/corneliussenf/DroughtForestMortalityEurope (with a permanent version under: https://doi.org/10.5281/zenodo.3924667).

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

## Acknowledgements

AB, AR, and CSZ acknowledge funding by the Bavarian Ministry of Science and the Arts in the context of the Bavarian Climate Research Network (BayKliF).

## Author contributions

C.S. developed the research idea, with input from A.B. and C.S.Z. A.B. provided pre-processed climate data. C.S. processed all data and performed all statistical analyses. C.S. wrote the paper with input and revision from A.B., C.S.Z., R.S., and A.R.

## Funding

## Competing interests

The authors declare no competing interests.
