## [Peer Review File · Nature Communications]

Reviewers' Comments:

Reviewer #1:

Remarks to the Author:

This ms. directly addresses the ecologically and societally important question of how forest mortality in Europe over the period 1987-2016 is related to drought. Using unique annual data (recently developed by these authors) on spatial patterns of forest mortality derived from Landsat satellite imagery, and comparing to spatio-temporal patterns of climatic water balance CWB, a consistent relationship was found of increased forest canopy mortality in association with more severe drought. Particularly interesting and important findings include: the identification and mapping of a number of mortality "hotspots" (not all of which have been previously reported in internationally-accessible peer-reviewed literature); and the determination of "(T)he relationship between water availability and mortality showed threshold behavior, with excess mortality increasing steeply when the climatic water balance fell below -1.6 standard deviations of its long-term average".

Overall this paper is cleanly written and clearly presented. Still, upon reading this ms. and additional review of the methods as further described in other related recent Senf et al. papers cited in this ms, I do have a variety of questions and comments to consider:

- 1) How exactly were excess mortality areas calculated for this ms.? Were these quartile categories used to characterize entire grid cells, and then used to calculate the precise cumulative area (ha.) of excess mortality? Or, were original 30m-pixel Landsat TM data classified into the quartile levels and then aggregated across all grid cells subject to threshold CWB drought stresses?
- 2) Given the 30m original resolution of the Landsat data, and also that as per Senf & Seidl 2020, "(T)he 25 million individual disturbance patches had a mean patch size of 1.09 ha (range between 1st and 99th percentile 0.18 – 10.10 ha); it seems a pity to then greatly coarsen the tree mortality data to match CRU climate data resolution of 0.5 degree cells. Are there really no higher-resolution gridded climatic data available for Europe, something comparable PRISM data in US? Or was computational tractability a consideration?
- 3) I was surprised that throughout this ms., "climate change" is noted but warming is never explicitly discussed, and that "drought" ("water availability", as defined by CWB-climatic water balance [which does include temperature effects in the PET term) is the only climatic parameter discussed – even though the role of warming temps is implicit and important to the emergence of greater drought stress and increasing levels of tree mortality from multiple mortality agents under climate change (as the authors of this ms. all know well, and as already reflected in the solid referencing in this ms). So, were any temp-related climatic analyses implemented, but not reported in this ms.? If not, it would be very interesting to consider the role of temp metrics, or other more-temperature-influenced drought metrics (e.g., climatic water deficit, as suggested elsewhere by co-author Zang, in Zang+ 2020). Even as-is, without any additional temperature-related analyses, the findings of this ms. might resonate more strongly if at least framed in a context where warming is explicitly linked to greater drought/water stress on trees during drought – as well as likely amplifying other tree-killing disturbance processes such as insect outbreaks, fires, windstorms, and direct physiological stress on trees. Such textual framing could be done very easily and concisely, without adding any additional references; one minimalist possibility is presented under "particular text comments", farther below.
- 4) Similarly, it seems like the nonlinear drought/tree-mortality relationships nicely demonstrated in this ms. easily could be concisely textually linked to recent literature that increasingly recognizes the importance of extreme-event drought/heat episodes as drivers of pulses of tree mortality (including relatively short-duration heat waves, like Europe summer 2003, SW Australia summer 2011).
- 5) Of course, an obvious question is how much of the mapped "canopy mortality" is due to harvest (in particular), as well as other mortality agents? While recognizing the challenges posed by the limitations described in the various recent Senf et al. papers that are working with these new canopy mortality data (e.g., missing pan-European reference data on disturbances, and the effects of management actions that are often overlaid atop natural disturbances [e.g., sanitation logging

atop tree dieback]) – still, perhaps it could be useful to try and separate and quantify the various mortality agents in some of your important hotspot areas at least, maybe by using local/regional records of harvest, fire, windthrow, etc. (?).

6) Relative to your closing sentence in “Results”: “Drought explained in total 6 % of the pan-European variability in forest canopy mortality”. Taken as a stand-alone statement, this doesn’t sound very impressive, and it would be useful context for readers for you to textually identify/describe the details of all other factors that “explain” the other 94% of the variability (or alternatively, to reference a supplementary table where those details are self-evident).

7) Why not address in this ms. the clear increasing trend in canopy mortality shown in Fig S2, likely correlated with increasing temperature (especially given that you already know that this increasing trend will be strengthened whenever analogous data for 2018 & 2019 are added, as these most recent years were big years for canopy mortality in Europe [with 2018 already identified as bigger than any other year in the 30-year preceding period in Senf, Sebald, Seidl 2020])? Are there any text comments you would be comfortable to make (even qualitatively) about the relative magnitude of 2018-2020 canopy mortality compared to your 30-year baseline, maybe even only for Central Europe ?

Also, here are a few particular text comments/suggestions, including some minor grammatical considerations:

PDF version, Line 24: consider alternative to “essential”, perhaps “important”.

In initial introductory paragraph, some possible ways to efficiently incorporate the importance of warming into the setup:

Line 28: could insert “hotter” before “droughts” and make plural, so “hotter droughts are...”

Line 33: again, insert “hotter”, to read “hotter drought is an inciting factor in...”

Line 36: insert “increasingly” before “susceptible”.

Line 51: substitute “across” for “in”, so “across Europe”.

Line 75: rearrange slightly, to : “Finally, we identified...”

Line 223: “For instance, we do not identify causal mortality agents..”

Line 225: revise to: “guarantees inclusion of all excess mortality...”

Line 404: fix reference 25, should be “Siwecki”, and “1998” (not 2007).

Review by Craig D. Allen, USGS.

Reviewer #2:

Remarks to the Author:

The paper performs a spatially and temporally broad quantification effort of mortality across Europe, and links this to deviation in cumulative water balance with a modelling framework. The authors find extensive mortality events confirming previous work, as well as potentially understudied events. The approach identifies -1.6 sigma as a threshold CWB for mortality. While the results are primarily confirmatory, the scale and temporal coverage make this a useful contribution likely of interest to a broad group of researchers. Overall I find the argument that a long term spatially extensive quantification of drought mortality is necessary to be compelling, though I note that certain of the co-authors appear to have previously published a similar analysis

over a shorter time period a few years ago. I find the work to be well-conducted and rigorous, but, this is challenging to evaluate from sparse presentation of the methods, and somewhat incomplete justification of the modeling framework.

MAJOR:

(1)

The self-referencing as explanation of methods (line 243, 260) is obscure. Recognizing that remote sensing is not my expertise, but that I am a researcher interested in tree mortality, it's hard to evaluate the methods of this paper. The authors cite their own preprint (not peer-reviewed?), which itself further cites other of their work in place of detailed explanation of the disturbance mapping methods. I am only willing to go "one level down" i.e. I looked at the preprint, but I am not willing to go look at the referenced articles etc. in that preprint. I find the manuscript does not provide enough information about the methods to answer my following questions:

-How are human disturbances and fire or drought related disturbances differentiated? It seems the preprint describes a human trained machine-learning approach, but that's all I could really tell in my (admittedly quick) reading. It would be good at least to report classification statistics of the method.

-In the manual classifications, what are the criteria for identifying specific events? IE how did the observers become able to produce a "well-informed call"?

-The "established modelling framework based on logistic regression" cited in 260 could be briefly described.

(2)

Line 284. Some issues with model presentation:

-Eqn numbers would be helpful (see next)

-In the third and fourth lines, the specification for the random slopes and intercepts: given the final model was Bayesian, should replace equals signs with "~" as these are priors.

-What is the subscript "m"?

-Perhaps note "MVN" denotes "multivariate normal".

-Please justify the use of the multivariate normal priors? Is this to reduce over-parameterization of the model? It would be good to report the effective number of parameters here to understand how well the MVN is constraining these parameters (from DIC Spiegelhalter, D. J., Best, N. G., Carlin, B. P., & Van Der Linde, A. (2002). Bayesian measures of model complexity and fit. *Journal of the royal statistical society: Series b (statistical methodology)*, 64(4), 583-639.

). Why not just construct a hierarchical model drawing spatially/temporally varying intercepts and effects from the global/population level parameters? This would allow estimation of the spatial/temporal variation (which appears small anyways from Table 1) without over-parameterization. The need for regularized (constrained) priors and exponential error distribution also may suggest issues with over-parameterization in (true) parameter identifiability.

-Why are intercepts (alphas) and slopes (betas) drawn from the covariance matrix given they are likely on very different scales? Since CWB is centered and scaled, alphas and betas should be uncorrelated anyways (and I see in table 1 neither of these covariance parameters is significant).

MINOR AND SUGGESTIONS

Line 41. Ref 24 is not Europe.

Line 52. I would revise to include something about the time-series length, as the only limitation of Neumann et al. 2017 is comparatively short time period.

Line 76. Regions and time periods? Q3 refers to "where and when"

Methods

Line 215. What is meant by "consistent estimate"? Suggest delete.

Line 225-227. This goes back to my comment on unclear methods. I have no idea what kind of

data is included in drought related disturbance and this sentence suggests logging is?
Line 227-232. Again, really difficult to evaluate these claims.

Reviewer #3:

Remarks to the Author:

This paper relates a previously developed forest mortality map with maps of a measure of drought stress across Europe. A clever analysis of this correlation is the strength of the paper, providing a template that may be useful for other studies focusing on links between observed forest trends and ancillary variables like drought stress. I especially appreciate the exploration of implications shown in Figures 1 and S1. Description of this analysis, though lengthy, is necessary and clear. Aside from some minor matters, detailed below, my questions regarding this manuscript relate to interpretation of the results. The Abstract claims that the paper provides evidence that "excess forest mortality (i.e., canopy mortality exceeding the long-term mortality trend) is significantly related to drought across continental Europe" and that "drought is an essential driver of tree mortality in Europe at the continental scale." However, the analysis shows that the fraction of mortality that can be linked to drought is 0.4 – 2.7% of the continent's total mortality over a 30-year period (line 148). While this interval does not include zero, and the use of the phrase "significantly related" is justified, I would not say that this is a large or surprising fraction of mortality. Figure 4 does make a compelling case that drought is an important mortality factor in at least some parts of Europe. The capacity to extend the analysis in this spatially explicit way is a real strength of the analysis. I am a little confused why the apparently strong dependence of mortality on drought occurring in Moldova and Ukraine went unmentioned and undiscussed. I am also surprised that fire was not more seriously discussed as a possible drought-dependent mechanism of mortality. The mortality maps used here did not distinguish the type of disturbance, and the analysis focused on gross mortality above average. Other secondary drought-related disturbance mechanisms were at least mentioned: wind-throw and sanitation cuts. Fire would seem to be the big one in some areas, and a quick search of Google Scholar suggests that fires in Ukraine may be increasing because of reduced emphasis on forest management and because of climate change. I believe that the Discussion would be more compelling with a comprehensive consideration of the role of fire, at least in locations where Figure 4 indicates the effect of drought on mortality is large.

Overall, I believe the authors have done an outstanding analysis, which they have described clearly. Addressing my questions regarding interpretation of results may require substantially revised presentation and discussion of the results.

Minor concerns

Table 1. It would be helpful to more explicitly describe the temporal and spatial variation terms. This phrase – "(i.e., variability in space and time)" – does not sufficiently link these variables to your study design.

Line 132. I think "the local average" could be more accurately described as the pixel's long-term average, right? Local average might imply some kind of spatial neighborhood function.

Line 204. "This indicates that Europe's forest ecosystems are generally well adapted..." This seems to me to be conjecture, so it might be better to use a word like "suggests" rather than "indicates"?

Line 220. "results are based on validated maps of canopy mortality and not – as often used – maps of simple vegetation indices" This is a misstatement of progress in the field of change detection. The authors cite an unrefereed preprint for all details of the process used for detecting mortality. This may be ok (I don't know this journal's policy for such citations), but my read of the cited paper indicates that the authors are aware that mapping of simple change in vegetation indices is far from the state of the art.

Line 344. The short section "Hotspots mapping and area-estimates" seems redundant with earlier details.

Reviewer #1

This ms. directly addresses the ecologically and societally important question of how forest mortality in Europe over the period 1987-2016 is related to drought. Using unique annual data (recently developed by these authors) on spatial patterns of forest mortality derived from Landsat satellite imagery, and comparing to spatio-temporal patterns of climatic water balance CWB, a consistent relationship was found of increased forest canopy mortality in association with more severe drought. Particularly interesting and important findings include: the identification and mapping of a number of mortality “hotspots” (not all of which have been previously reported in internationally-accessible peer-reviewed literature); and the determination of “(T)he relationship between water availability and mortality showed threshold behavior, with excess mortality increasing steeply when the climatic water balance fell below -1.6 standard deviations of its long-term average”.

Overall this paper is cleanly written and clearly presented. Still, upon reading this ms. and additional review of the methods as further described in other related recent Senf et al. papers cited in this ms, I do have a variety of questions and comments to consider:

Response: We thank the reviewer for his thorough and informative review! We answered all questions and comments in detail below.

1) How exactly were excess mortality areas calculated for this ms.? Were these quartile categories used to characterize entire grid cells, and then used to calculate the precise cumulative area (ha.) of excess mortality? Or, were original 30m-pixel Landsat TM data classified into the quartile levels and then aggregated across all grid cells subject to threshold CWB drought stresses?

Response: We thank the reviewer for this question and the opportunity to clarify how excess mortality was calculated. In essence, we use the 30 m mortality map to calculate annual mortality rates (proportion of forest area that dies each year) for each 0.5° grid cell. We then fit a trend line for each 0.5° grid cell’s time series of annual mortality rates (see Figure S2 for one grid cell). From the trend line, we then calculate the residuals as measure of excess mortality (i.e., more mortality than expected under the long-term trend) and mortality deficit (i.e., less mortality as expected). The residuals were subsequently centered by dividing them through the fitted values, resulting in a fractional deviation from the long-term trend. Hence – to answer the question – we first aggregate the 30 m maps to the 0.5° grid and then calculated excess mortality. To clarify those important methodological details for the reader, we extended the methods description significantly, adding more details on how the data was processed and how excess mortality was calculated (L. 272ff):

“We identified anomalies in tree mortality from an existing high-resolution canopy mortality map created from Landsat satellite data³⁰ and being available under the following link: <https://doi.org/10.5281/zenodo.3924381>. While the data is described in detail in Senf and Seidl 2020³⁰, we here provide the salient details necessary for understanding our approach: The map has a spatial grain of 30 m and is based on a supervised classification of Landsat satellite image time series. The map indicates for each pixel whether a canopy mortality event has occurred between 1987 and 2016 and if so, in which year it occurred. The formal agent of the canopy mortality event (i.e., natural or human cause) is not classified. The map also identifies all pixels that have been forested over the same time period. The overall accuracy of the map is 92 %, with a commission error of 15 % and an omission error of 33 % for

detecting canopy mortality events³⁰. Consequently, the map is conservative, omitting some disturbances in favor of not predicting false canopy mortality events.

We spatially overlaid the canopy mortality map with a 0.5-degree grid (total of 3,113 grid cells) and spatially aggregated the disturbance maps to the annual sum of 30 m pixels indicating canopy mortality (i.e., the annual total area of canopy mortality recorded within a 0.5-degree grid cell). We also aggregated the total forest area to the 0.5° grid cell level, allowing us to derive an annual mortality rate per grid cell (total annual area of canopy mortality divided by the total forest area per grid cell(?)). We excluded all cells that consisted mostly of non-forest ecoregions according to Olsen et al.⁵² (e.g., tundra and grassland ecoregions in northern Scandinavia), which resulted in the exclusion of 200 grid cells (6 %). Based on previous research we expected the average mortality rate to increase across Europe^{53,54}. However, in some years there can be substantially higher mortality rates than what would be expected even under a long-term increasing trend, a fact that we here define as excess mortality. To quantify excess mortality for each year and grid cell, we used an established modeling framework based on logistic regression (described in Senf et al. 2018⁵³) to model each cell's long-term trend in canopy mortality. In essence, the model predicts the annual proportion of pixels with canopy mortality over the total number of forested pixels using a logistic model with binomial error distribution and time as predictor, modeling the change in the average proportion of forest area subject to canopy mortality (i.e., the mortality rate) over time (see trend line in Supplementary Figure S3). We subsequently derived the residuals from the model and normalized them by the annual fitted values from the trend line. The resultant value indicates the annual fractional deviation in mortality from the long-term trend: A value close to zero indicates no change in mortality compared to the long-term trend, values greater than zero indicate excess mortality, and values smaller than zero indicate a mortality deficit (see Supplementary Materials Figure S1).”

2) Given the 30m original resolution of the Landsat data, and also that as per Senf & Seidl 2020, “(T)he 25 million individual disturbance patches had a mean patch size of 1.09 ha (range between 1st and 99th percentile 0.18 – 10.10 ha); it seems a pity to then greatly coarsen the tree mortality data to match CRU climate data resolution of 0.5 degree cells. Are there really no higher-resolution gridded climatic data available for Europe, something comparable PRISM data in US? Or was computational tractability a consideration?”

Response: A very good comment and good questions. First of all, there are only limited reanalysis datasets with at least monthly resolution for all of Europe. While there are higher-resolution climate datasets and some national datasets, our analysis required a consistent dataset for all European countries (even outside the EU, e.g., Albania, Ukraine, Moldova, Belarus are often missing as they are not member-states of the EU). Second, we need at least some aggregation level to calculate excess mortality. That is, while doing analyses at the original 30-m resolution is highly interesting (e.g., analyzing spatial patterns), there needs to be a coarser spatial level for calculating the expected annual mortality rate and subsequently the excess mortality per grid cell(?) used as response variable in this manuscript (see previous answer). Computational limitations were not a limiting factor in this study (even though the processing is very intense).

3) I was surprised that throughout this ms., “climate change” is noted but warming is never explicitly discussed, and that “drought” (“water availability”, as defined by CWB-climatic

water balance [which does include temperature effects in the PET term) is the only climatic parameter discussed – even though the role of warming temps is implicit and important to the emergence of greater drought stress and increasing levels of tree mortality from multiple mortality agents under climate change (as the authors of this ms. all know well, and as already reflected in the solid referencing in this ms). So, were any temp-related climatic analyses implemented, but not reported in this ms.? If not, it would be very interesting to consider the role of temp metrics, or other more-temperature-influenced drought metrics (e.g., climatic water deficit, as suggested elsewhere by co-author Zang, in Zang+ 2020). Even as-is, without any additional temperature-related analyses, the findings of this ms. might resonate more strongly if at least framed in a context where warming is explicitly linked to greater drought/water stress on trees during drought – as well as likely amplifying other tree-killing disturbance processes such as insect outbreaks, fires, windstorms, and direct physiological stress on trees. Such textual framing could be done very easily and concisely, without adding any additional references; one minimalist possibility is presented under “particular text comments”, farther below.

Response: We thank the reviewer for this thoughtful comment. We agree that changes in temperature, and not only precipitation, are important for explaining tree mortality under drought. While we used the climatic water balance as predictor (which indirectly includes temperature via evapotranspiration, as pointed out by the reviewer), we also did test other indices in the initial development of this research. For instance, we ran the same models using VPD (i.e., another indicator driven by water availability *and* temperature) and obtained very similar results for the statistical model, with only minor differences in the hotspot maps and area estimates. In order to make the importance of warming more explicit in the manuscript, we followed the reviewer’s suggestions outlined in the detailed comments below. Furthermore, we added a new paragraph on the effect of drought on secondary mortality agents (see answer to comment #5). We also added a note in the final paragraph that increasing hotter droughts might be particularly challenging for the future of Europe’s forests (L. 265):

“These limitations notwithstanding, we here provide the first systematic continental-scale evidence that drought is an important agent of tree mortality in Europe. Our results suggest that an increase in the frequency and severity of drought under climate change, that is an increase of hotter drought³⁶, could lead to substantial tree mortality across Europe’s forests, as has been already observed in 2018 and 2019^{45,51}.”

We also revised the discussion to better reflect the role of increasing temperature and hotter droughts for the risk of future drought-related forest diebacks (L. 209):

“It also suggests that under increasing drought intensity³⁸ and hotter droughts under climate change³⁶, tree mortality could increase disproportionately. In this respect we note that the effect of drought on tree mortality identified here was strongest for droughts lasting over several months (March to July), suggesting that an increase in drought duration – as expected for droughts under climate change³⁹ – might further amplify tree mortality in the future.”

4) Similarly, it seems like the nonlinear drought/tree-mortality relationships nicely demonstrated in this ms. easily could be concisely textually linked to recent literature that increasingly recognizes the importance of extreme-event drought/heat episodes as drivers of pulses of tree mortality (including relatively short-duration heat waves, like Europe summer

2003, SW Australia summer 2011).

Response: We agree and revised the text accordingly (L. 207): “The threshold behavior emerging from our data is well in line with previous research and current process-understanding of tree responses to drought^{36,37}. It also suggests that under increasing drought intensity³⁸ and hotter droughts under climate change³⁶, tree mortality could increase disproportionately.”

5) Of course, an obvious question is how much of the mapped “canopy mortality” is due to harvest (in particular), as well as other mortality agents? While recognizing the challenges posed by the limitations described in the various recent Senf et al. papers that are working with these new canopy mortality data (e.g., missing pan-European reference data on disturbances, and the effects of management actions that are often overlaid atop natural disturbances [e.g., sanitation logging atop tree dieback]) – still, perhaps it could be useful to try and separate and quantify the various mortality agents in some of your important hotspot areas at least, maybe by using local/regional records of harvest, fire, windthrow, etc. (?).

Response: We fully agree that it is an obvious caveat of our analysis that we do not separate formal agents of mortality. While using excess mortality addresses this problem to some degree (i.e., it is, for instance, very unlikely to have large excess mortality caused by regular timber harvest), there still can be several agents that ultimately kill a tree. While we work on developing new methods for separating formal agents with our canopy mortality maps (and some good process has been made recently), attributing agents to satellite-based changes in forest canopy is still a big challenge. We nonetheless agree that the manuscript lacked some detailed discussion on potential formal agents of mortality. We hence revised the manuscript by adding a new paragraph to the discussion that discusses potential agents of change and also bringing examples from regions where we have better knowledge and/or data. We also added a new analysis to the supplement (Figure S12), showing that fire is a strong driver of drought-related mortality in Europe. The new paragraph starts in line 215 and reads as follows:

“While droughts in general and hotter droughts in particular can trigger mortality directly, it is most likely the combined effect of drought and secondary mortality agents that lead to the tree mortality hotspots identified in this study. One of the most important mortality agents in terms of hotter drought is fire¹²⁻¹⁵. For example, the year 2005 marks one of the most severe fire seasons in Portugal and Spain in recent decades⁴⁰ (excluding the even more severe year 2017 not included in our analysis), explaining the significant hotspot of drought-related tree mortality identified in our analysis in the same year across the south-west of the Iberian peninsula (Figure 2). Likewise, Ukraine and neighboring countries were hit by widespread forest fires during the drought of 2007 and 2011/2012, as were the Balkan peninsula in 2000 and the Baltic states including Belarus in 2002⁴¹, all of which are well visible in our hotspot assessment (Figure 2). Overall, our estimates of drought related excess mortality correlate reasonably well with the total area burned reported by countries (Pearson $r = 0.43$ [0.33 – 0.52]; Supplementary Figure S12), indicating that a large proportion of the drought-related excess mortality identified in this study may be caused by fire disturbances. Hotter droughts also are an inciting factor of tree mortality from insects⁶⁻¹¹. For example, during the 2003 heatwave in Europe, mortality from several insect species increased in direct response to the increasing heat and lack of water in France¹¹, supporting the widespread dieback identified in our analysis (see Figure 2 for year 2003). However, we like to note that while bark beetle mortality (mostly by *Ips typographus*) was linked to drought in past studies^{11,42-44}, there is

often a temporal lag of one to three years between drought occurrence and the detection of dead trees in satellite products^{14,42}. Hence, our analysis likely does not include mortality caused by bark beetles, which contributed significantly to tree mortality in recent years across Central Europe⁴⁵. In contrast to indirect mortality by fire or insects, there are also well-documented instances of tree dieback as direct consequence of drought. For example, in Spain in 1994 widespread dieback of Oak and Pine was caused by drought²¹, despite a low fire season (Supplementary Figure S12) (see Figure 3). While we recorded a peak in mortality in 1994 in Figure 3, the area affected was highly uncertain, mostly because drought-related dieback can be more spatially(?) diffuse than mortality caused by fire or insects and thus more difficult to detect using satellite data⁴⁶.”

6) Relative to your closing sentence in “Results”: “Drought explained in total 6 % of the pan-European variability in forest canopy mortality”. Taken as a stand-alone statement, this doesn’t sound very impressive, and it would be useful context for readers for you to textually identify/describe the details of all other factors that “explain” the other 94% of the variability (or alternatively, to reference a supplementary table where those details are self-evident).

Response: We thank the reviewer for this comment. While we agree that 6% does not sound impressive, we’d like to highlight that this results from a simple global model with a single predictor. Given the scale of our analysis, we believe that it is indeed surprising that water availability alone explains already 6% of all excess mortality occurring in Europe! The remaining variability is unexplained in our model, that is, it is residual variance. The residual variance results from both pulses of mortality unrelated to CWB (i.e., wind) and stochasticity. We revised the text in the manuscript to clarify this point (L. 88):

“The final model was substantially better than a null model based solely on random spatial and temporal variability (Table S1) and explained 6 % of the pan-European variability in forest canopy mortality.”

7) Why not address in this ms. the clear increasing trend in canopy mortality shown in Fig S2, likely correlated with increasing temperature (especially given that you already know that this increasing trend will be strengthened whenever analogous data for 2018 & 2019 are added, as these most recent years were big years for canopy mortality in Europe [with 2018 already identified as bigger than any other year in the 30-year preceding period in Senf, Sebald, Seidl 2020])? Are there any text comments you would be comfortable to make (even qualitatively) about the relative magnitude of 2018-2020 canopy mortality compared to your 30-year baseline, maybe even only for Central Europe?

Response: A very good comment, though we’re deliberately careful in making speculations about the drivers underlying the general increase in canopy mortality observed for most countries in Europe. The observed increases in canopy mortality can have a variety of reasons, including both increasing mortality resulting from increasing temperatures (especially facilitating bark beetle outbreaks in Central Europe), but also increasing utilization of forest resources. For example, from personal correspondence and observations we know that the strong increase in canopy mortality in Slovenia observed in Senf, Sebald and Seidl 2020 results from increased windthrow and bark beetle in recent years. In fact, most of the harvest in Slovenia is now salvage logging. There are other countries, however, that also significantly increased the harvesting intensity during the past 30 years, which also

contributes to the increase in canopy loss observed from space. In a previous paper (Senf et al. 2018) we showed that trends in canopy mortality correlate with both increases in temperature but also increases in growing stock across Central Europe. It is thus very difficult to disentangle the underlying root causes that drive the general increase. Due to this speculative nature of the relationship between the increases in temperature and general increases in canopy mortality, we refrained from adding additional text to the manuscript.

We agree though that the years 2018 to 2020 were likely outside the recent range of variability in terms of canopy mortality, most likely due to drought-triggered intense bark beetle outbreaks occurring in Central Europe. We note this also in the manuscript (“[...] and we suggest that an increase in the frequency and severity of drought under climate change, that is an increase of hotter drought³⁶, could lead to substantial tree mortality across Europe’s forests, as has been already observed in 2018 and 2019^{45,51}.”), but we yet lack reliable spatial data for the most recent years to include in our analysis. We’re working on an update of our disturbance maps until 2020, with a particular focus on bark beetle, but this work is still in progress and will necessarily have to wait until 2021 to be completed.

PDF version, Line 24: consider alternative to “essential”, perhaps “important”.

Response: Agreed and changed as suggested.

Line 28: could insert “hotter” before “droughts” and make plural, so “hotter droughts are...”

Response: Agreed and changed as suggested.

Line 33: again, insert “hotter”, to read “hotter drought is an inciting factor in...”

Response: Agreed and changed as suggested.

Line 36: insert “increasingly” before “susceptible”.

Response: Agreed and changed as suggested.

Line 51: substitute “across” for “in”, so “across Europe”.

Response: Agreed and changed as suggested.

Line 75: rearrange slightly, to: “Finally, we identified...”

Response: Agreed and changed as suggested.

Line 223: “For instance, we do not identify causal mortality agents...”

Response: Agreed and changed as suggested.

Line 225: revise to: “guarantees inclusion of all excess mortality...”

Response: Agreed and changed as suggested.

Line 404: fix reference 25, should be “Siwecki”, and “1998” (not 2007).

Response: Good catch, thank you! Changed as suggested.

Reviewer #2

The paper performs a spatially and temporally broad quantification effort of mortality across Europe, and links this to deviation in cumulative water balance with a modelling framework. The authors find extensive mortality events confirming previous work, as well as potentially understudied events. The approach identifies -1.6 sigma as a threshold CWB for mortality. While the results are primarily confirmatory, the scale and temporal coverage make this a useful contribution likely of interest to a broad group of researchers. Overall I find the argument that a long term spatially extensive quantification of drought mortality is necessary to be compelling, though I note that certain of the co-authors appear to have previously published a similar analysis over a shorter time period a few years ago. I find the work to be well-conducted and rigorous, but, this is challenging to evaluate from sparse presentation of the methods, and somewhat incomplete justification of the modeling framework.

Response: We thank the reviewer for this informative review. We especially acknowledge the critical evaluation of our methods, which greatly helped to improve their description in the revised version of the manuscript.

The self-referencing as explanation of methods (line 243, 260) is obscure. Recognizing that remote sensing is not my expertise, but that I am a researcher interested in tree mortality, it's hard to evaluate the methods of this paper. The authors cite their own preprint (not peer-reviewed?), which itself further cites other of their work in place of detailed explanation of the disturbance mapping methods. I am only willing to go "one level down" i.e. I looked at the preprint, but I am not willing to go look at the referenced articles etc. in that preprint. I find the manuscript does not provide enough information about the methods to answer my following questions:

Response: We thank the reviewer very much for this important comment. We agree that methods should be explained in a way that makes them traceable even without reading through all references provided in the text. We note, though, that the key reference explaining the spatially explicit disturbance maps has undergone extensive peer review and has, in the meantime, been accepted for publication in *Nature Sustainability*. We nonetheless took the comments and suggestions of the reviewer very serious and have revised and extended our methods description to improve the description of our approach and make the methods as self-contained as possible. Please see our detailed answer to the following comments and questions.

How are human disturbances and fire or drought related disturbances differentiated? It seems the preprint describes a human trained machine-learning approach, but that's all I could really tell in my (admittedly quick) reading. It would be good at least to report classification statistics of the method.

Response: Thanks for this comment, we agree that there were some details missing in the methods description. First of all, the canopy mortality map used in this study does not separate causal agents of change, it just indicates loss of forest canopy. Second, we fully agree that classification accuracy statistics are important for judging the quality of the maps. We revised the methods description to include both a note that the map does not distinguish

between formal agents of change, and report the classification accuracies of the models used (L. 273ff):

“We identified anomalies in tree mortality from an existing high-resolution canopy mortality map created from Landsat satellite data³⁰ and being available under the following link: <https://doi.org/10.5281/zenodo.3924381>. While the data is described in detail in Senf and Seidl 2020³⁰, we here provide the salient details necessary for understanding our approach: The map has a spatial grain of 30 m and is based on a supervised classification of Landsat satellite image time series. The map indicates for each pixel whether a canopy mortality event has occurred between 1987 and 2016 and if so, in which year it occurred. The formal agent of the canopy mortality event (i.e., natural or human cause) is not classified. The map also identifies all pixels that have been forested over the same time period. The overall accuracy of the map is 92 %, with a commission error of 15 % and an omission error of 33 % for detecting canopy mortality events³⁰. Consequently, the map is conservative, omitting some disturbances in favor of not predicting false canopy mortality events.”

In the manual classifications, what are the criteria for identifying specific events? IE how did the observers become able to produce a “well-informed call”?

Response: A good question which we are happy to answer, even though it is not directly related to the methods of this manuscript. The collection of reference data for training the mapping algorithm was based on the manual interpretation of both Landsat image trajectories and Landsat image chips, as well as high-resolution imagery. The interpretation is based on protocols and software developed by Cohen et al in 2010, which are well-established tools for assessing changes in the tree canopy across large spatial and long temporal scales. Using image interpretation of satellite data allows us to assess changes in canopy mortality for areas that are challenging to cover by field surveys and where historic data is simply not available (e.g., large parts of eastern Europe). The methods are all well-established and have been extensively validated (Cohen et al. 2010 and Cohen et al. 2017) and applied across a large range of forest ecosystems (Pflugmacher et al. 2012, Potapov et al. 2015, Hermosilla et al. 2015, Senf et al. 2015, Cohen et al. 2016, Senf et al. 2018). Given the spatial and temporal scale of the disturbance maps presented in Senf and Seidl 2020 and used in our analysis, we highlight that there is – to the best of our knowledge – no better approach for creating reference data than using manual interpretation of satellite time series. That being said, whenever humans are involved there will be errors. This applies to the manual interpretation of satellite images as well as to any other ecological measurements (like measuring the height of trees, counting beetles, etc.). In order to ensure the quality of our assessment we performed a series of checks to bolster our confidence in the data, all of which are outlined in detail in our previous publications. In short, we first validated each interpreter against a reference interpretation done by the two most experienced interpreters. This internal cross-check highlighted a very high consistency (from 85% to 100%) among interpreters. Nonetheless, there were some inconsistencies for plots of very low disturbance severity, that is plots where the spectral change was small in comparison to the overall noise in the time series. To make the final call on those plots consistently across the data (and thus avoid bias in the analysis), the most experienced interpreters re-evaluated all of those plots. While this second round of interpretation doesn’t guarantee that the decision made by the interpreter is correct, it however guarantees that the decision is consistent among plots.

Cohen, W. B. et al. Forest disturbance across the conterminous United States from 1985–2012: The emerging dominance of forest decline. *Forest Ecology and Management* **360**, 242–252 (2016).

Cohen, W. B., Yang, Z. & Kennedy, R. Detecting trends in forest disturbance and recovery using yearly Landsat time series: 2. TimeSync — Tools for calibration and validation. *Remote Sensing of Environment* **114**, 2911–2924 (2010).

Cohen, W. et al. How Similar Are Forest Disturbance Maps Derived from Different Landsat Time Series Algorithms? *Forests* **8**, 98 (2017).

Hermosilla, T., Wulder, M. A., White, J. C., Coops, N. C. & Hobart, G. W. Regional detection, characterization, and attribution of annual forest change from 1984 to 2012 using Landsat-derived time-series metrics. *Remote Sensing of Environment* **170**, 121–132 (2015).

Pflugmacher, D., Cohen, W. B. & E. Kennedy, R. Using Landsat-derived disturbance history (1972–2010) to predict current forest structure. *Remote Sensing of Environment* **122**, 146–165 (2012).

Potapov, P. V. et al. Eastern Europe’s forest cover dynamics from 1985 to 2012 quantified from the full Landsat archive. *Remote Sensing of Environment* **159**, 28–43 (2015).

Senf C, Pflugmacher D, Wulder MA, Hostert P (2015) Characterizing spectral–temporal patterns of defoliator and bark beetle disturbances using Landsat time series. *Remote Sensing of Environment* **170**:166–177.

Senf, C. et al. Canopy mortality has doubled across Europe’s temperate forests in the last three decades. *Nature Communications* **9**, 4978 (2018).

The “established modelling framework based on logistic regression” cited in 260 could be briefly described.

Response: We agree with the reviewer’s suggestion and added a brief description of the modelling framework (L. 296):

“To quantify excess mortality for each year and grid cell, we used an established modeling framework based on logistic regression (described in Senf et al. 2018⁵³) to model each cell’s long-term trend in canopy mortality. In essence, the model predicts the annual proportion of pixels with canopy mortality over the total number of forested pixels using a logistic model with binomial error distribution and time as predictor, modeling the change in the average proportion of forest area subject to canopy mortality (i.e., the mortality rate) over time (see trend line in Supplementary Figure S3).”

Line 284. Some issues with model presentation:

Response: We thank the reviewer for the thorough feedback on our modelling framework. We tried to be as transparent on the statistical analysis as possible and the detailed feedback by the reviewer confirmed our expectations that reporting all statistical models in detail (and

not just verbally describing the analysis) is helpful for making statistical analyses robust and reproduceable. Please see detailed answer to the following comments and questions.

Eqn numbers would be helpful (see next)

Response: We apologize for not using equation numbers in the initial manuscript. Equation numbers were added to the revised manuscript.

In the third and fourth lines, the specification for the random slopes and intercepts: given the final model was Bayesian, should replace equals signs with “~” as these are priors.

Response: We thank the reviewer for this comment, but we like to note that the specifications for the random slope and intercept (eq. 1.2 and 1.3 in revised manuscript) are not priors, but part of the model likelihood. Nonetheless, the reviewer is correct that given the expression as probability distributions the tilde operator is the correct convention. We revised the formulas.

What is the subscript “m”?

Response: The subscript m relates to the response variable, indicating that both the residual error and residual variance relate to the response variable. To clarify this point, we substituted m by *mort*, making it more distinct from the other single-letter subscripts used for indexing space (i) and time (t).

Perhaps note “MVN” denotes “multivariate normal”.

Response: Agreed and noted in L. 290.

Please justify the use of the multivariate normal priors? Is this to reduce over-parameterization of the model? It would be good to report the effective number of parameters here to understand how well the MVN is constraining these parameters (from DIC Spiegelhalter, D. J., Best, N. G., Carlin, B. P., & Van Der Linde, A. (2002). Bayesian measures of model complexity and fit. *Journal of the royal statistical society: Series b (statistical methodology)*, 64(4), 583-639.). Why not just construct a hierarchical model drawing spatially/temporally varying intercepts and effects from the global/population level parameters? This would allow estimation of the spatial/temporal variation (which appears small anyways from Table 1) without over-parameterization. The need for regularized (constrained) priors and exponential error distribution also may suggest issues with over-parameterization in (true) parameter identifiability.

Response: We thank the reviewer for this comment. As note before, those are not priors, but the MVN used for describing the random slope and intercept are part of the model likelihood. Using a MVN for modeling random intercept and slope jointly is common practice in multilevel-modelling as outlined in several recent statistical textbooks (e.g., Gelman and Hill 2007, p. 279, McElreath 2020, Chapter 13). Also, common frequentist R-packages for hierarchical modeling (e.g., *lme4*) use a MVN definition of random intercepts and slopes, as do R-packages used for Bayesian modeling (e.g., *rstanarm*, *brms*). The reason for using a

MVN is to allow a correlation between random intercept and slope (which happens quite often, see for example Sorensen et al. 2016), but the correlation can also be zero, in which case it just turns into independent normal distributions for both the random slope and intercept. As hyper-prior on the variance-covariance matrix, which is the only parameter that needs a prior (see eq. 1.2 and 1.3), the standard implementation of *brms* was used (as noted in the text). The standard implementation in *brms* follows Monnahan et al. 2016 and uses a Cholesky decomposition to convert the variance-covariance matrix into a correlation matrix and then uses a LKJ-prior, as recommend in both the *Stan* manual (https://mc-stan.org/docs/2_18/stan-users-guide/multivariate-hierarchical-priors-section.html) and in recent statistical textbooks (e.g., McElreath 2020, Chapter 13), but see also the fabulous manual by Sorensen et al. 2016.

Second, we like to note that the formulation as hierarchical (i.e. partial-pooled) model in fact avoids over-parameterization, as the only parameters to be estimated (and thus with a prior) are the parameters of the variance-covariance matrix! This is the great advantage of hierarchical models, as the deviation of each grid-cell/year from the global estimate is not a parameter by itself (as would be with, e.g., interaction terms), but they are random draws from the MVN.

Regarding the comment “*Why not just construct a hierarchical model drawing spatially/temporally varying intercepts and effects from the global/population level parameters? This would allow estimation of the spatial/temporal variation (which appears small anyways from Table 1) without over-parameterization*”, we believe that there was a misunderstanding due to a suboptimal description of the methods in the initial submission. The aim was to test whether including the ‘fixed’ effect (CWB) improves the model. We hence compared a null model (without CWB) to a model containing CWB as predictor (see Table S1). The null model includes an intercept with random variation of the intercept among grid-cells and years. The model thus draws randomly from the spatial and temporal variability in the intercept, which – and here the reviewer is correct – does not stem from Σ_i and Σ_t , as there is no random variation in slopes in the null model, but from normal distributions with simple variance parameters. In essence, the null model is a standard random intercept model with two crossed random effects. We revised the description of the model accordingly:

“We finally compared three model set-ups: First, we fitted a null-model containing only an intercept but the similar random effect structure as the models described above. That is, the intercept was allowed to vary randomly among grid cells and years following $\alpha_i \sim N(0, \sigma_{\alpha_i}^2)$ and $\alpha_t \sim N(0, \sigma_{\alpha_t}^2)$. The null-model thus assumes that anomalies in tree mortality emerge from pure stochasticity in time and space.”

We hope that the revision clarifies all questions.

McElreath R (2020) *Statistical Rethinking: A Bayesian Course with Examples in R and Stan*. Version 2. Chapman and Hall/CRC.

Monnahan CC, Thorson JT, Branch TA (2017) Faster estimation of Bayesian models in ecology using Hamiltonian Monte Carlo. *Methods in Ecology and Evolution* 8:339–348.

Gelman, A., & Hill, J. (2006). *Data analysis using regression and multilevel/hierarchical models*. Cambridge University Press.

Sorensen, T., Hohenstein, S., & Vasishth, S (2016) Bayesian linear mixed models using Stan: A tutorial for psychologists, linguists, and cognitive scientists, *The Quantitative Methods for Psychology*, 12(3):175-200.

Why are intercepts (alphas) and slopes (betas) drawn from the covariance matrix given they are likely on very different scales? Since CWB is centered and scaled, alphas and betas should be uncorrelated anyways (and I see in table 1 neither of these covariance parameters is significant).

Response: Thanks for this comment. The *brms* package uses a reparameterization as described in Monnahan et al. 2016 via a Cholesky decomposition of the covariance matrix into a correlation matrix, which then can be centered and scaled using the mean and variance of the individual parameters. The great advantage of this method is that indeed the final priors can all be centered on zero with a unit standard deviation, which greatly improves the MCMC sampling. While those specific details of the statistical methods used in our analysis are indeed very interesting, we did not mention them in the manuscript as they have been well documented in recent statistical textbooks (e.g., McElreath 2020, Chapter 13) and are the de-facto standard in modern Bayesian modeling.

McElreath R (2020) *Statistical Rethinking: A Bayesian Course with Examples in R and Stan*. Version 2. Chapman and Hall/CRC.

Monnahan CC, Thorson JT, Branch TA (2017) Faster estimation of Bayesian models in ecology using Hamiltonian Monte Carlo. *Methods in Ecology and Evolution* 8:339–348.

Line 41. Ref 24 is not Europe.

Response: Good catch! Should have been Bigler et al. 2006. Citation updated accordingly, also in discussion section.

Line 52. I would revise to include something about the time-series length, as the only limitation of Neumann et al. 2017 is comparatively short time period.

Response: Agreed and revised to: “We here present a first systematic and quantitative analysis of drought-related tree mortality over 30 years and across all of continental Europe’s forests.”

Line 76. Regions and time periods? Q3 refers to “where and when”

Response: Agreed and revised to: “Finally, we identified hotspots of drought-related canopy mortality, that is regions and years where excess canopy mortality co-occurred with drought, [...]”

Line 215. What is meant by “consistent estimate”? Suggest delete.

Response: Agreed and revised as suggested.

Line 225-227. This goes back to my comment on unclear methods. I have no idea what kind of data is included in drought related disturbance and this sentence suggests logging is?

Response: As outlined in the previous comment and now properly described in the revised methods description, the initial canopy mortality maps do include all potential agents of forest canopy loss.

Line 227-232. Again, really difficult to evaluate these claims.

Response: Please see answer to previous comment.

Reviewer #3

This paper relates a previously developed forest mortality map with maps of a measure of drought stress across Europe. A clever analysis of this correlation is the strength of the paper, providing a template that may be useful for other studies focusing on links between observed forest trends and ancillary variables like drought stress. I especially appreciate the exploration of implications shown in Figures 1 and S1. Description of this analysis, though lengthy, is necessary and clear.

Response: We thank the reviewer for the positive and detailed review!

Aside from some minor matters, detailed below, my questions regarding this manuscript relate to interpretation of the results. The Abstract claims that the paper provides evidence that “excess forest mortality (i.e., canopy mortality exceeding the long-term mortality trend) is significantly related to drought across continental Europe” and that “drought is an essential driver of tree mortality in Europe at the continental scale.” However, the analysis shows that the fraction of mortality that can be linked to drought is 0.4 – 2.7% of the continent’s total mortality over a 30-year period (line 148). While this interval does not include zero, and the use of the phrase “significantly related” is justified, I would not say that this is a large or surprising fraction of mortality. Figure 4 does make a compelling case that drought is an important mortality factor in at least some parts of Europe. The capacity to extend the analysis in this spatially explicit way is a real strength of the analysis. I am a little confused why the apparently strong dependence of mortality on drought occurring in Moldova and Ukraine went unmentioned and undiscussed.

Response: We thank the reviewer for this comment. We agree that the amount of drought-related mortality was not surprising. We thus revised the abstract and conclusion to be more careful with the interpretation of our results:

Abstract: “Pulses of tree mortality caused by drought have been reported recently in forests around the globe, but large-scale quantitative evidence is lacking for Europe. Analyzing high-resolution annual satellite-based canopy mortality maps from 1987 to 2016 we here show that excess forest mortality (i.e., canopy mortality exceeding the long-term mortality trend) is significantly related to drought across continental Europe. The relationship between water availability and mortality showed threshold behavior, with excess mortality increasing steeply when the integrated climatic water balance from March to July fell below -1.6 standard deviations of its long-term average. For -3.0 standard deviations the probability of excess canopy mortality was 91.6% (83.8 – 97.5%). Overall, drought caused approximately 500,000 ha of excess forest mortality between 1987 and 2016 in Europe. We here provide evidence that drought is an important driver of tree mortality at the continental scale, and suggest that a future increase in drought could trigger widespread tree mortality in Europe.”

We also agree with the reviewer that the spatial analysis is a big advantage of our analysis, thank you for acknowledging this! Regarding the large mortality pulse in Ukraine and Moldova, we revised the discussion, but please see the answer to the next comment for details.

I am also surprised that fire was not more seriously discussed as a possible drought-dependent mechanism of mortality. The mortality maps used here did not distinguish the type of

disturbance, and the analysis focused on gross mortality above average. Other secondary drought-related disturbance mechanisms were at least mentioned: wind-throw and sanitation cuts. Fire would seem to be the big one in some areas, and a quick search of Google Scholar suggests that fires in Ukraine may be increasing because of reduced emphasis on forest management and because of climate change. I believe that the Discussion would be more compelling with a comprehensive consideration of the role of fire, at least in locations where Figure 4 indicates the effect of drought on mortality is large.

Overall, I believe the authors have done an outstanding analysis, which they have described clearly. Addressing my questions regarding interpretation of results may require substantially revised presentation and discussion of the results.

Response: We thank the reviewer very much for this comment. We agree that we did not well discuss the role of fire in drought-related tree mortality. The reviewer also is correct that there was a lack of discussion with respect to the large mortality events in Ukraine and Moldova, which are indeed related to fire. To improve the manuscript in respect to this comment, we added an additional paragraph to the discussion section, specifically discussing potential causal agents of tree death linked to drought. In this paragraph, we also bring several examples of known fire and insect events that are linked to drought, including examples from Ukraine. To support the discussion, we moreover added a new analysis, comparing the drought-related excess mortality reported in our manuscript to official fire reports by the EU (see Supplementary Figure S12). The new paragraph reads as follows (L. 215ff):

“While droughts in general and hotter droughts in particular can trigger mortality directly, it is most likely the combined effect of drought and secondary mortality agents that lead to the tree mortality hotspots identified in this study. One of the most important mortality agents in terms of hotter drought is fire¹²⁻¹⁵. For example, the year 2005 marks one of the most severe fire seasons in Portugal and Spain in recent decades⁴⁰ (excluding the even more severe year 2017 not included in our analysis), explaining the significant hotspot of drought-related tree mortality identified in our analysis in the same year across the south-west of the Iberian peninsula (Figure 2). Likewise, Ukraine and neighboring countries were hit by widespread forest fires during the drought of 2007 and 2011/2012, as were the Balkan peninsula in 2000 and the Baltic states including Belarus in 2002⁴¹, all of which are well visible in our hotspot assessment (Figure 2). Overall, our estimates of drought related excess mortality correlate reasonably well with the total area burned reported by countries (Pearson $r = 0.43$ [0.33 – 0.52]; Supplementary Figure S12), indicating that a large proportion of the drought-related excess mortality identified in this study may be caused by fire disturbances. Hotter droughts also are an inciting factor of tree mortality from insects⁶⁻¹¹. For example, during the 2003 heatwave in Europe, mortality from several insect species increased in direct response to the increasing heat and lack of water in France¹¹, supporting the widespread dieback identified in our analysis (see Figure 2 for year 2003). However, we like to note that while bark beetle mortality (mostly by *Ips typographus*) was linked to drought in past studies^{11,42-44}, there is often a temporal lag of one to three years between drought occurrence and the detection of dead trees in satellite products^{14,42}. Hence, our analysis likely does not include drought-related mortality by bark beetles, which contributed significantly to tree mortality in recent years across Central Europe⁴⁵. In contrast to indirect mortality by fire or insects, there are also well-documented instances of tree dieback as direct consequence of drought. For example, in Spain in 1994 widespread dieback of Oak and Pine was caused by drought²¹, despite a low fire season (Supplementary Figure S12) (see Figure 3). While we recorded a peak in mortality in 1994 in Figure 3, the area affected was highly uncertain, mostly because drought-related dieback can be more diffuse than mortality caused by fire or insects and thus more difficult to detect using satellite data⁴⁶.”

Table 1. It would be helpful to more explicitly describe the temporal and spatial variation terms. This phrase – “(i.e., variability in space and time)” – does not sufficiently link these variables to your study design.

Response: Agreed and revised to better explain the spatial and temporal variability and how it related to our study design: “. Parameters are grouped into smooth term parameters (i.e., degree of smoothing), group-level parameters (i.e., variability of both intercept and slope in space [i.e., grid cells i] and time [i.e., years t]), population-level parameters (i.e., population averages after accounting for spatial and temporal variability), and family-specific parameters.”

Line 132. I think “the local average” could be more accurately described as the pixel’s long-term average, right? Local average might imply some kind of spatial neighborhood function.

Response: Agreed and revised as suggested.

Line 204. “This indicates that Europe’s forest ecosystems are generally well adapted...” This seems to me to be conjecture, so it might be better to use a word like “suggests” rather than “indicates”?

Response: Agreed and revised to “suggested”.

Line 220. “results are based on validated maps of canopy mortality and not – as often used – maps of simple vegetation indices” This is a misstatement of progress in the field of change detection. The authors cite an unrefereed preprint for all details of the process used for detecting mortality. This may be ok (I don’t know this journal’s policy for such citations), but my read of the cited paper indicates that the authors are aware that mapping of simple change in vegetation indices is far from the state of the art.

Response: We thank the reviewer for this comment. First of all, we note that the preprint cited in the manuscript has undergone extensive peer-review and has, in the meantime, been accepted for publication by *Nature Sustainability* (doi: 10.1038/s41893-020-00609-y). Second, our comment related to studies that correlate climate variability with changes in vegetation indices (such as the NDVI), without validating what a change in the vegetation index actually means. A drop in NDVI during a drought year does not mean that drought has caused tree mortality, it could be simply due to a reduction in photosynthetic activity or loss of leaf area that is restored in the following year. Our maps, however, depict true mortality events, that is a loss of the tree canopy, and have been verified by roughly 20,000 reference pixels manually interpreted as described in detail in the respective reference. While we agree that mapping tree loss from satellite data is not novel per se, there is still a lack of fine-scale products that (i) are validated, (ii) cover large areas such as all of Europe, (iii) cover long time periods (i.e., multiple decades), and (iv) are consistent through time. The Hansen global forest cover loss map, for instance, is a great global product, but only starts in 2001, is temporally inconsistent (see: <https://data.globalforestwatch.org/datasets/14228e6347c44f5691572169e9e107ad>) and the accuracy for specific regions is unknown. For those reasons, there is a lack of studies linking

drought to tree mortality (and not just changes in NDVI or similar spectral indices) at continental scales; a research gap we aim to fill with our manuscript. To clarify this point in our manuscript we have revised the text to read: “Moreover, our results are based on validated maps of canopy mortality and not – as often used – maps of anomalies in vegetation indices without classification into distinct mortality events^{39,40}. Our results thus report actual mortality events and not only changes in photosynthetic activity, which might be ephemeral.”.

Line 344. The short section “Hotspots mapping and area-estimates” seems redundant with earlier details.

Response: While we agree that there is some redundancy with the previous sections, we believe that there is additional information (e.g., how the hotspots were mapped and how uncertainty was quantified) that is needed for fully understanding our methods and results. We thus decided to keep the section.

Reviewers' Comments:

Reviewer #1:

Remarks to the Author:

I am satisfied with the thorough responses to all of the review comments, and with the associated revisions to the manuscript.

Craig D. Allen

Reviewer #2:

Remarks to the Author:

I find the authors revisions and responses to my previous comments to be clear and compelling. Given the majority of my original comments were simply issues with clarity or requests for more significant documentation of the studies' methodology, and that the authors have addressed all of these with changes, I have no further comments upon this manuscript.

Reviewer #3:

Remarks to the Author:

I am satisfied with revisions related to my earlier review comments. I think the paper is now in a form appropriate for publication by the journal.